# Association between serum soluble (pro)renin receptor level and worsening of cardiac function in hemodialysis patients: A prospective observational study

Yoshifumi Amari[1,2], Satoshi Morimoto[2]*, Takeshi Iida[1,2], Hiroki Takimoto[1], Hidenobu Okuda[1], Takatomi Yurugi[1], Yasuo Oyama[3], Naoki Aoyama[4], Fumitaka Nakajima[5], Atsuhiro Ichihara[2]

1 Department of Nephrology, Moriguchi Keijinkai Hospital, Osaka, Japan, 2 Department of Endocrinology and Hypertension, Tokyo Women's Medical University, Tokyo, Japan, 3 Department of Nephrology and Dialysis, Neyagawa Keijinkai Clinic, Osaka, Japan, 4 Department of Nephrology and Dialysis, Moriguchi Keijinkai Clinic, Osaka, Japan, 5 Department of Nephrology and Dialysis, Kadoma Keijinkai Clinic, Osaka, Japan

* morimoto.satoshi@twmu.ac.jp

**Data Availability Statement:** All relevant data are within the paper and its Supporting Information files.

## Abstract

The (pro)renin receptor ((P)RR) is cleaved to generate soluble (P)RR (s(P)RR), which reflects the status of the tissue renin-angiotensin system. Hemodialysis (HD) patients have a poor prognosis due to the increased prevalence of cardiovascular diseases. The present study aimed to investigate whether serum s(P)RR level is associated with the worsening of cardiac function in HD patients. A total of 258 maintenance HD patients were recruited and serum s(P)RR concentration was measured. Background factors in patients who survived (S group) and patients who died (D group) during the 12-month follow-up period and relationships between serum s(P)RR level and changes in cardiac function during the follow-up period in the S group were investigated. The median serum s(P)RR value at baseline was 29.8 ng/ml. Twenty-four patients died during the follow-up period. Cardiothoracic ratio, human atrial natriuretic peptide (hANP), brain natriuretic peptide (BNP), and E over e-prime were significantly higher in the D group. In the S group, changes in hANP or BNP were significantly greater in the higher serum s(P)RR group than in the lower serum s(P)RR group. High serum s(P)RR level was significantly correlated with changes in BNP, independent of other factors. High serum s(P)RR level was associated with increases in BNP, independent of other risk factors, suggesting that an increased expression of (P)RR may be associated with a progression of heart failure in HD patients and that serum s(P)RR concentration could be used as a biomarker for selecting patients requiring intensive care.

## Introduction

The (pro)renin receptor ((P)RR) consists of 350 amino acids with a single transmembrane domain and binds preferentially to renin and prorenin [1]. The binding of prorenin to the

**Funding:** The authors received no specific funding for this work.

**Competing interests:** The authors have declared that no competing interests exist.

extracellular domain of the (P)RR induces non-proteolytic renin activation [2], which accelerates the conversion of angiotensinogen to angiotensin (Ang) I. This process plays a key role in the regulation of the tissue renin-angiotensin system (RAS) [1]. (P)RR is cleaved by processing enzymes to generate soluble (P)RR (s(P)RR), which is secreted into the extracellular space and found in blood. These findings suggest that s(P)RR can serve as a biomarker reflecting the status of the tissue RAS and activity of (P)RR [3, 4].

Hemodialysis (HD) patients have a poor prognosis due to an increased prevalence of cardiovascular disease (CVD) [5, 6]. It has been reported that patients with heart failure had significantly higher plasma s(P)RR levels than control subjects [7]. We have previously reported that serum s(P)RR level is associated with arteriosclerosis, independent of other risk factors in HD patients [8]. These data prompted us to hypothesize that blood s(P)RR level could be associated with progression of CVD. However, it remains undermined if serum s(P)RR level is associated with the changes in indices of cardiovascular dysfunction. On the basis of these background findings, the present study aimed to investigate the relationship between serum s(P)RR level and changes in background factors including cardiac function and atherogenic factors.

## Materials and methods

### Study subjects

The participants were outpatients on maintenance HD at Kadoma Keijinnkai Clinic, Neyagawa Keijinnkai Clinic, and Moriguchi Keijinnkai Clinic in Osaka Prefecture, Japan. All three clinics are affiliated with Moriguchi Keijinkai Hospital, Osaka, Japan. This study was approved by the ethical committee of Tokyo Women's Medical University (approval number: 2703), and all patients were enrolled after obtaining written informed consent. A total of 258 maintenance HD patients who could be followed up for 12 months were recruited consecutively between March and May 2013.

### Background factors

At the start of this study, we collected information on the study population, including age, sex, body mass index (BMI), primary disease (diabetic or not), duration of HD, smoking status, selected medication, CTR, and Kt/V. BMI was calculated as follows: BMI = *post-dialysis value of body weight (kg) / [height (m)]$^2$} × 100*. Post-dialysis cardiothoracic ratio (CTR) values were obtained on the first dialysis day of the week. The Kt/V was calculated on the 1st dialysis day of the week using the following equation, the formula of Daugirdas [9]: Kt/V = *—Ln [post-dialysis value of BUN / pre-dialysis value of BUN—0.008 x dialysis time] + (4–3.5 x post-dialysis value of BUN / pre-dialysis value of BUN) x amount of drainage / post-dialysis body weight}*

### Blood examinations

Non-fasting blood samples were taken while patients were lying in bed in a supine position after at least 15 minutes of rest on the first dialysis day of the week. The following pre-dialysis parameters were measured: hemoglobin (Hb), high-density lipoprotein cholesterol (HDL-C), low-density lipoprotein cholesterol (LDL-C), triglyceride (TG), albumin-corrected calcium (Ca), inorganic phosphorus (IP), intact parathyroid hormone (Intact-PTH), creatinine (Cre), uric acid (UA), C-reactive protein (CRP), and albumin (Alb) levels.

The following post-dialysis values were measured by conventional methods at an external testing laboratory (Kishimoto, Inc., Tomakomai City, Japan):human atrial natriuretic peptide

(hANP), a marker of body fluid volume [10–12] and brain natriuretic peptide (BNP), a marker of left ventricular dysfunction [13].

Pre-dialysis serum s(P)RR levels were measured using an enzyme-linked immunosorbent assay (ELISA) kit (Takara Bio Inc., Otsu City, Japan) consisting of a solid-phase sandwich ELISA with antibodies highly specific for each protein [14].

## Physiological function tests

**Echocardiography.** Echocardiography was performed on a non-dialysis day as previously described using the Vivid S6 System (GE Healthcare, Milwaukee, WI, USA), and cardiac functions as follows were estimated: 1) left ventricular ejection fraction (LVEF), a marker of contractile activity; 2) interventricular septum thickness (IVST), posterior LV wall thickness (PWT), and left ventricular mass index (LVMI), markers of cardiac hypertrophy [15]; and 3) E over e-prime (E/e') and deceleration time (Dec-T), markers of left ventricular diastolic function [16].

*Ankle-brachial index and brachial ankle pulse wave velocity (baPWV).* The ankle-brachial index (ABI) values (average and lower values) and baPWV values (average and higher values) were measured on a non-dialysis day using a volume-plethysmographic apparatus PWV/ABI (Omron Healthcare Co., Ltd., Kyoto, Japan) following previously described methods [17]. BaPWV cannot be estimated properly when the ABI is less than 0.9 because arterial occlusion retards baPWV [18, 19]. Therefore, patients with ABI <0.9 were excluded from the analysis of baPWV.

## Computed tomography (CT)

Body fat distribution was determined using CT imaging on a non-dialysis day with the use of a 64-row multislice CT scanner (Aquilion 64; Toshiba, Tokyo, Japan). The subcutaneous fat area (SFA) and visceral fat area (VFA) were measured at the level of the umbilicus using Ziostation 2 software (Ziosoft, Tokyo, Japan).

## Study protocols

Pre-dialysis serum s(P)RR levels on the first dialysis day of the week were measured at the start of the study. The patients were divided into two groups (higher and lower groups) according to serum s(P)RR values, and we compared the groups in terms of background factors, blood data, physiological function data, and CT data. Patients were followed up for 12 months or until death from any cause. We compared baseline characteristics in the survival and death groups. In the survival group, we compared background factors, blood data, physiological function data, and CT data at baseline and after 12 months, and the changes in these values (Δ) were compared in the higher and lower serum s(P)RR groups. Finally, we examined the association between serum s(P)RR level and Δ BNP.

## Statistical analyses

For all continuous variables, parametric variables and non-parametric variables were expressed as mean ± SD and median with interquartile ranges (25th and 75th percentiles), respectively. Intergroup comparisons of parameters were performed using a Wilcoxon signed-rank test or Mann-Whitney U test. Categorical variables were presented as the number of patients and compared using a chi-square test. We calculated the Spearman's rank correlation coefficient between Δ BNP and background factors, blood data, physiological function data, and CT data. Multiple regression analyses were performed using factors that showed

significant correlation with Δ BNP as independent variables. The level of significance was defined as P<0.05. All analyses were performed using Bell Curve for Excel (Social Survey Research Information Co. Ltd., Tokyo, Japan).

## Results

### Characteristics of the study patients at baseline

The median serum s(P)RR value at baseline was 29.8 ng/ml. Table 1 details the baseline characteristics of the study patients in the two groups (higher and lower serum s(P)RR groups) and includes background factors, blood data, physiological function data, and CT data. The number of patients with measurements of ABI and baPWV was 247. Forty-four of these patients showed an ABI of <0.9. As a result, a total of 203 patients were included in the baPWV analysis. CTR, Hb, TG, and CRP levels were significantly higher, and IVST was significantly lower in the higher serum s(P)RR group than in the lower serum s(P)RR group. With respect to antihypertensive treatment, the ratio of patients taking RAS-inhibitors (RAS-Is), such as angiotensin-converting enzyme inhibitors, angiotensin receptor blockers (ARBs), and renin inhibitors, was significantly higher in the lower serum s(P)RR group (83.7%) than in the higher serum s(P)RR group (69.0%). Primary disease (diabetic or not) was not significantly different between the 2 groups. In addition, neither BMI nor Kt/V was significantly different between the 2 groups, suggesting that hemodialysis parameters may not have strong influences on the serum s(P)RR levels.

### Comparison of study patient characteristics at baseline in the survival and death groups

During the 12-month follow-up period, one patient received a renal transplant and 24 deaths (9.3%) were recorded. There were 14 cardiovascular deaths (5.4%), including one due to acute myocardial infarction, six to congestive heart failure, five to lethal arrhythmia, and two to sudden unexpected deaths. Ten non-cardiovascular deaths (3.9%) were recorded, including five because of infectious diseases; two, cachexia; and three, cancer. The 1-year survival rate was 90.7%. Table 2 details the comparison of study patient characteristics at baseline in the survival and death groups. Age, male sex, CTR, CRP, hANP, BNP, E/e', and low ABI (<0.9) were significantly higher, and BMI, HDL-C, Cre, Alb, and LVEF were significantly lower in the death group than in the survival group. No significant differences in serum s(P)RR levels were noted between patients in the death group (29.8 ± 6.1 ng/ml) and the survival group (29.9 [26.6–34.1] ng/ml). No significant differences in primary disease (DM/no DM) was noted between in the survival and death groups. With respect to antihypertensive treatment, the ratio of patients taking RAS-Is and calcium channel blockers (CCBs) were significantly higher in the survival group (79.4% and 77.3%, respectively) than the death group (45.8% and 41.7%, respectively).

### Characteristics of the study patients at baseline and after 12 months in the survival group

In patients who survived, CTR, Ca, CRP, hANP, and BNP values, IVST, PWT, LVMI, Dec-T, average ABI, lower ABI, average baPWV, and higher baPWV were significantly higher and LVEF, E/e', SFA, and VFA were significantly lower after 12 months than at baseline (Table 3). Δ hANP and Δ BNP values and Δ IVST were significantly greater in the higher serum s(P)RR group than in the lower group (Table 4, Fig 1). By contrast, Δ Hb, Δ Cre, and UA values were significantly lower in the higher serum s(P)RR group than in the lower group (Table 4).

**Table 1. Comparison of characteristics of the study subjects at baseline between higher and lower groups of serum s(P)RR concentration.**

| | higher group | lower group | P |
|---|---|---|---|
| | [s(P)RR $\geq$29.8 ng/ml] | [s(P)RR <29.8 ng/ml] | |
| | (n = 129) | (n = 129) | |
| **Background factors** | | | |
| Age (y.o.) | 72 (62–77) | 68 (59–74) | 0.070 |
| Gender (male/female) | 71/58 | 75/54 | 0.615 |
| Body mass index (kg/m²) | 21.8 (19.9–24) | 21.6 (19.4–24.2) | 0.549 |
| Primary disease (DM/non DM) | 55/74 | 68/61 | 0.105 |
| Duration of hemodialysis therapy (months) | 48 (24–84) | 48 (24–72) | 0.788 |
| Smoking history (yes/no) | 55/74 | 60/69 | 0.531 |
| Medications (yes/no) | | | |
| RAS-I | 89 / 40 | 108 / 21 | 0.005 |
| β-blocker | 44 / 85 | 54 / 75 | 0.217 |
| CCB | 90 / 39 | 100 / 29 | 0.188 |
| CTR (%) | 53.0 (49.4–55.4) | 51.7 ± 5.2 | 0.040 |
| Kt/V | 1.36 ± 0.26 | 1.34 ± 0.22 | 0.313 |
| **Blood tests** | | | |
| Hemoglobin (g/dl) | 10.9 ± 1.0 | 10.7 (10.2–11.2) | 0.036 |
| HDL cholesterol (mg/dl) | 46.0 (38.0–55.0) | 45.0 (36.0–55.0) | 0.632 |
| LDL cholesterol (mg/dl) | 91.3 ± 30.5 | 83.0 (64.0–103.0) | 0.203 |
| Triglyceride (mg/dl) | 96.0 (68.0–136.0) | 81.0 (55.0–113.0) | 0.004 |
| Calcium (mg/dl) | 8.8 (8.5–9.2) | 8.8 (8.6–9.1) | 0.970 |
| Inorganic phosphorus (mg/dl) | 5.4 (4.4–6.0) | 5.1 ± 1.1 | 0.064 |
| Intact-Parathyroid hormone (pg/ml) | 139.0 (84.0–185.0) | 123.0 (76.0–176.0) | 0.305 |
| Creatinine (mg/dl) | 9.5 ± 2.5 | 9.7 ± 2.6 | 0.478 |
| Uric acid (mg/dl) | 7.3 ± 1.4 | 7.0 ± 1.3 | 0.053 |
| CRP (mg/dl) | 0.13 (0.05–0.49) | 0.07 (0.05–0.19) | <0.001 |
| Albumin (g/dl) | 3.7 ± 0.3 | 3.7 (3.5–3.9) | 0.422 |
| hANP (pg/ml) | 48.2 (28.2–78.2) | 46.7 (31.5–74.9) | 0.866 |
| BNP (pg/ml) | 136.6 (76.4–287.4) | 148.9 (74.9–244.5) | 0.974 |
| **Physical function tests** | | | |
| Echocardiography | | | |
| LVEF (%) | 67.6 (62.8–72.1) | 67.2 (63.5–71.5) | 0.361 |
| IVST (mm) | 12.0 (11.1–13.5) | 12.4 (11.4–13.5) | 0.036 |
| PWT (mm) | 12.0 (11.0–13.0) | 12.0 (11.3–13.1) | 0.090 |
| LVMI (g/m²) | 171.1 (144.9–202.7) | 180.5 ± 50.0 | 0.202 |
| E/e' | 18.0 (14.0–22.1) | 18.8 (14.9–22.2) | 0.404 |
| Dec-T | 229.0 (199.8–264.1) | 231.7 ± 54.1 | 0.817 |
| ABI | | | |
| $\geq$0.9 / <0.9 | 96/25 | 107/19 | 0.252 |
| average value | 1.15 (1.05–1.23) | 1.17 (1.07–1.24) | 0.378 |
| lower value | 1.13 (0.95–1.20) | 1.13 (1.00–1.21) | 0.776 |
| baPWV (cm/s) | | | |
| average value | 1794.8 (1568.4–2065.9) | 1853.5 (1606.8–2176.8) | 0.719 |
| higher value | 1907.0 (1639.5–2165.8) | 1921.0 (1641.5–2234.0) | 0.774 |
| **Abdominal CT** | | | |
| Subcutaneous fat area (cm²) | 110.9 (72.2–171.2) | 105.5 (59.8–155.6) | 0.105 |

(*Continued*)

**Table 1.** (Continued)

| | higher group | lower group | P |
|---|---|---|---|
| | [s(P)RR ≥29.8 ng/ml] | [s(P)RR <29.8 ng/ml] | |
| | (n = 129) | (n = 129) | |
| Visceral fat area (cm$^2$) | 79.1 (47.7–119.3) | 69.4 (33.1–112.1) | 0.097 |

DM, diabetes mellitus; RAS-I, renin-angiotensin system inhibitor; CCB, calcium channel blocker; CTR, cardiothoracic ratio; HDL cholesterol, high-density lipoprotein cholesterol; LDL cholesterol, low-density lipoprotein cholesterol; CRP, C-reactive protein; hANP, human atrial natriuretic peptide; BNP, brain natriuretic peptide; LVEF, left ventricular ejection fraction; IVST, interventricular septum thickness; PWT, posterior left ventricular wall thickness; LVMI, left ventricular mass index; E/e', E over e-prime; Dec-T, deceleration time; ABI, ankle-brachial index; baPWV, brachial-ankle pulse wave velocity

### Relationship between Δ BNP and background factors in the survival group

In patients who survived, LDL-C and TG levels were significantly and positively correlated, and hANP and BNP levels were significantly and negatively correlated with Δ BNP (Table 5). The ratio of patients taking RAS-Is, β-blockers, and CCBs did not show significant relationship with ΔBNP. Serum s(P)RR level was not associated with ΔBNP level (Table 5). On the other hand, when the subjects were divided into the higher and lower s(P)RR groups, ΔBNP was significantly higher in the higher s(P)RR group than in the lower s(P)RR group (Fig 1). Multiple regression analyses testing higher serum s(P)RR (≥29.8 ng/ml), LDL-C level, TG, hANP, and BNP as independent variables revealed that higher serum s(P)RR (≥29.8 ng/ml) correlated with Δ BNP, independent of other factors (Table 6).

## Discussion

The present study aimed to investigate the relationship between high serum s(P)RR level and changes in cardiac function and atherogenic factors and demonstrated three major findings. Firstly, some types of atherogenic factors and cardiac functions were significantly worse in the higher serum s(P)RR group than in the lower group. Secondly, in patients who survived, Δ hANP and Δ BNP levels, and Δ IVST were significantly greater in the higher serum s(P)RR group than in the lower group. Finally, in patients who survived, the association between high Δ BNP and higher serum s(P)RR was observed even after correction for other Δ BNP-related factors. These data suggest that an increased expression of (P)RR may be associated with a progression of heart failure in HD patients.

### Serum s(P)RR levels

It has been shown that basal blood s(P)RR levels differ among reports with a tendency of blood s(P)RR levels to be high in older subjects [4, 20], patients with heart failure [7], renal dysfunction [21], or Graves' disease [22], and pregnant women [23, 24]. The reason of the alterations of these values may be due to the differences of the disease and conditions of the subjects [24]. Our study subjects undergoing maintenance HD also showed high serum s(P)RR levels, which were significantly higher when compared with subjects with normal renal function [8]. Although the reason for this remains unclear, there are several possible explanations. First, elevated serum s(P)RR levels may reflect an increase in expression of (P)RR in the kidneys. Second, increased expression of (P)RR in other organs such as the heart and brain, where the expression levels of (P)RR are extremely high [1], may be the cause of elevated serum s(P)RR level. Third, elevated serum s(P)RR levels may be simply due to a decrease or loss of excretion from the kidney.

**Table 2. Comparison of characteristics of the study subjects at baseline between in survival and death groups.**

| | survival | death | P |
|---|---|---|---|
| **Background factors** | | | |
| n | 233 | 24 | |
| Age (y.o.) | 68 (60–76) | 77 ± 10 | <0.001 |
| Gender (male/female) | 129 / 104 | 16 / 8 | 0.002 |
| Body mass index (kg/m²) | 21.8 (19.8–24.1) | 20.5 ± 3.6 | 0.017 |
| Primary disease (DM/non DM) | 107 /126 | 16 / 8 | 0.053 |
| Duration of hemodialysis therapy (months) | 48 (24–84) | 39 (24–68) | 0.426 |
| Smoking history (yes/no) | 106 / 127 | 8 /16 | 0.254 |
| Medications (yes/no) | | | |
| RAS-I (yes/no) | 185 / 48 | 11 / 13 | <0.001 |
| β-blocker (yes/no) | 90 / 143 | 8 / 16 | 0.611 |
| CCB (yes/no) | 180 / 53 | 10 / 14 | <0.001 |
| CTR (%) | 51.1 (49.0–55.0) | 56.0 ± 5.8 | <0.001 |
| Kt/V | 1.35±0.24 | 1.34±0.19 | 0.421 |
| **Blood tests** | | | |
| s(P)RR (ng/ml) | 29.9 (26.6–34.1) | 29.8 ± 6.1 | 0.795 |
| Hemoglobin (g/dl) | 10.8 (10.3–11.4) | 10.6 ± 1.2 | 0.403 |
| HDL cholesterol (mg/dl) | 46.0 (37.0–56.0) | 40.8 ± 7.7 | 0.034 |
| LDL cholesterol (mg/dl) | 85.0 (66.0–108.0) | 90.7 ± 28.7 | 0.615 |
| Triglyceride (mg/dl) | 88.0 (59.0–126.0) | 97.7 ± 45.7 | 0.906 |
| Calcium (mg/dl) | 8.8 (8.5–9.2) | 8.8 ± 0.5 | 0.848 |
| Inorganic phosphorus (mg/dl) | 5.1 (4.3–5.8) | 5.4 (4.4–6.2) | 0.641 |
| Intact-Parathyroid hormone (pg/ml) | 129.0 (82.0–176.0) | 165.5 ± 102.5 | 0.385 |
| Creatinine (mg/dl) | 9.7 ± 2.6 | 8.3 ± 1.8 | 0.002 |
| Uric acid (mg/dl) | 7.2 ± 1.3 | 7.0 ± 1.4 | 0.567 |
| CRP (mg/dl) | 0.09 (0.05–0.23) | 0.53 (0.22–1.06) | <0.001 |
| Albumin (g/dl) | 3.7 (3.5–3.9) | 3.4 ± 0.4 | <0.001 |
| hANP (pg/ml) | 45.6 (28.7–71.7) | 68.6 (54.9–167.2) | <0.001 |
| BNP (pg/ml) | 136.4 (73.8–242.9) | 275.9 (185.0–961.6) | <0.001 |
| **Physical function tests** | | | |
| Echocardiography | | | |
| LVEF (%) | 67.6 (62.8–72.1) | 61.1 (41.9–65.2) | 0.001 |
| IVST (mm) | 12.0 (11.1–13.5) | 12.3 ±1.8 | 0.881 |
| PWT (mm) | 12.0 (11.0–13.0) | 11.9 (11.1–12.8) | 0.752 |
| LVMI (g/m²) | 171.1 (144.9–202.7) | 191.5 ± 44.7 | 0.092 |
| E/e' | 18.0 (14.0–22.1) | 24.2 ± 12.1 | 0.015 |
| Dec-T | 229.0 (199.8–264.1) | 215.3 ± 77.1 | 0.068 |
| ABI | | | |
| ≥0.9 / <0.9 | 187 / 40 | 16 / 7 | <0.001 |
| average value | 1.16 (1.06–1.23) | 1.12 ± 0.26 | 0.968 |
| lower value | 1.13 (0.98–1.21) | 1.11 (0.94–1.26) | 0.876 |
| baPWV (cm/s) | | | |
| average value | 1835.0 (1597.0–2156.3) | 1912.5 ± 421.5 | 0.565 |
| higher value | 1903.5 (1635.0–2208.0) | 2018.3 ± 407.6 | 0.295 |
| **Abdominal CT** | | | |
| Subcutaneous fat area (cm²) | 108.7 (69.7–162.7) | 92.3 ± 71.5 | 0.050 |

(*Continued*)

**Table 2.** (Continued)

|  | survival | death | P |
|---|---|---|---|
| Visceral fat area (cm$^2$) | 77.1 (41.9–120.4) | 61.0 (39.2–79.4) | 0.075 |

DM, diabetes mellitus; RAS-I, renin-angiotensin system inhibitor; CCB, calcium channel blocker; CTR, cardiothoracic ratio; s(P)RR, soluble (pro)renin receptor; HDL cholesterol, high-density lipoprotein cholesterol; LDL cholesterol, low-density lipoprotein cholesterol; CRP, C-reactive protein; hANP, human atrial natriuretic peptide; BNP, brain natriuretic peptide; LVEF, left ventricular ejection fraction; IVST, interventricular septum thickness; PWT, posterior left ventricular wall thickness; LVMI, left ventricular mass index; E/e', E over e-prime; Dec-T, deceleration time; ABI, ankle-brachial index; baPWV, brachial-ankle pulse wave velocity

## Comparison of characteristics of the study subjects at baseline between higher and lower groups of serum s(P)RR concentration

We have previously reported that serum s(P)RR level was significantly higher in patients with ABI <0.9 than in those with ABI ≥0.9, and this association was observed even after correction for atherogenic factors such as age, history of smoking, HbA1c, and LDL-C [8]. It was considered that high serum s(P)RR levels in HD patients may be associated with severe atherosclerosis of the lower limbs, independent of other risk factors, and that serum s(P)RR concentration could be used as a marker for atherosclerotic conditions in these patients [8].

The present study showed that atherogenic factors and markers of cardiac dysfunction were greater in the higher serum s(P)RR group than in the lower serum s(P)RR group (Table 1) in accordance with our previous report [8]. All these findings raise the possibility that an increased expression of tissue RAS and/or elevated serum s(P)RR concentration may be associated with a progression of CVD.

In our study, the patients in the lower serum s(P)RR group took RAS-Is more frequently than those in the higher serum s(P)RR group (Table 1). It has been reported that a significantly lower level of serum s(P)RR was observed in CKD patients treated with ARBs [21], consistent with our study.

## Comparison of characteristics of the study subjects at baseline in between survival and death groups

There is a significant association among inflammatory state, atherosclerotic CVD, and malnutrition, which has been described as malnutrition, inflammation, and atherosclerosis (MIA) syndrome. The presence of MIA syndrome is strongly associated with mortality in pre-dialysis chronic kidney disease and dialysis populations [25]. In line with these reports, our study demonstrated that CRP level was significantly higher, and BMI, Cre, and Alb were significantly lower in the death group than in the survival group (Table 2).

Manifestations of left ventricular disease are frequent and persistent, and are associated with high risks of heart failure and death in HD patients [26]. Heart failure was a strong, independent, adverse prognostic indicator in these patients [27]. In accordance with these findings, our study showed that CTR, hANP and BNP levels, and E/e' were significantly higher and LVEF was significantly lower in the death group than in the survival group (Table 2), suggesting that the status of systolic and diastolic dysfunction and heart failure are related to increased risk of death in HD patients. In addition, it has been reported that HD patients have advanced atherosclerosis, which is causally related to poor prognosis [5, 6], and we have previously reported that serum s(P)RR level was associated with arteriosclerosis, independent of other risk factors in these patients [8]. Therefore, we had anticipated that high serum s(P)RR level may be associated with poor prognosis. However, unexpectedly, in the present study, no

**Table 3. Characteristics of the study subjects at baseline and after 12 months in patients who survived.**

| | baseline | after 12 months | P | Δ |
|---|---|---|---|---|
| **Background factors** | | | | |
| Body mass index (kg/m$^2$) | 21.8 (19.8–24.1) | 21.7 (19.6–24.3) | 0.074 | 0.0 (-0.6–0.5) |
| CTR (%) | 51.1 (49.0–55.0) | 53.5 (49.5–57.7) | < 0.001 | 1.8 (-0.5–4.0) |
| **Blood tests** | | | | |
| Hemoglobin (g/dl) | 10.8 (10.3–11.4) | 10.9 (10.3–11.5) | 0.151 | 0.2 (-0.8–0.9) |
| HDL cholesterol (mg/dl) | 46.0 (37.0–56.0) | 46.0 (38.0–54.0) | 0.250 | 0.0 (-5.0–4.0) |
| LDL cholesterol (mg/dl) | 85.0 (66.0–108.0) | 87.6 ± 28.6 | 0.400 | 0.0 (-13.0–12.8) |
| Triglyceride (mg/dl) | 88.0 (59.0–126.0) | 84.0 (63.0–121.0) | 0.278 | 0.0 (-23.0–22.0) |
| Calcium (mg/dl) | 8.8 (8.5–9.2) | 9.0 ± 0.5 | <0.001 | 0.1 (-0.1–0.5) |
| Inorganic phosphorus (mg/dl) | 5.1 (4.3–5.8) | 5.1 (4.5–5.9) | 0.100 | 0.1 (-0.8–1.0) |
| Intact-Parathyroid hormone (pg/ml) | 129.0 (82.0–176.0) | 132.0 (85.0–184.0) | 0.375 | 2.0 (-44.0–52.0) |
| Creatinine (mg/dl) | 9.7 ± 2.6 | 10.1 ± 2.7 | 0.076 | 0.3 (-0.5–1.3) |
| Uric acid (mg/dl) | 7.2 ± 1.3 | 7.1 (6.3–7.9) | 0.066 | -0.1 (-0.8–0.6) |
| CRP (mg/dl) | 0.09 (0.05–0.23) | 0.10 (0.05–0.28) | 0.040 | 0.0 (0.0–0.1) |
| Albumin (g/dl) | 3.7 (3.5–3.9) | 3.7 (3.5–3.9) | 0.093 | 0.0 (-0.1–0.1) |
| hANP (pg/ml) | 45.6 (28.7–71.7) | 54.6 (33.4–76.0) | 0.003 | 2.5 (-11.7–23.4) |
| BNP (pg/ml) | 136.4 (73.8–242.9) | 184.7 (90.4–351.6) | <0.001 | 5.5 (-58.2–80.7) |
| **Physical function tests** | | | | |
| Echocardiography | | | | |
| LVEF (%) | 67.2 (62.0–71.7) | 65.1 (59.5–71.1) | <0.001 | -1.4 (-6.7–3.8) |
| IVST (mm) | 12.0 (11.2–13.5) | 12.4 (11.5–13.8) | <0.001 | 0.2 (-0.7–1.5) |
| PWT (mm) | 12.0 (11.1–13.0) | 12.2 (11.3–13.5) | <0.001 | 0.3 ± 1.4 |
| LVMI (g/m$^2$) | 172.7 (146.6–204.7) | 176.9 (152.1–219.2) | <0.001 | 1.8 (-20.1–37.7) |
| E/e' | 18.6 (14.0–22.6) | 18.1 (14.5–23.4) | <0.001 | 0.5 (-2.3–4.1) |
| Dec-T | 226.9 (196.1–264.1) | 231.0 (193.5–260.4) | 0.003 | 7.9 (-34.4–39.2) |
| ABI | | | | |
| average value | 1.16 (1.06–1.23) | 1.18 (1.05–1.25) | 0.005 | -0.1 (-0.1–0.1) |
| lower value | 1.13 (0.98–1.21) | 1.14 (0.97–1.22) | 0.001 | 0.0 (-0.1–0.1) |
| baPWV (cm/s) | | | | |
| average value | 1835.0 (1597.0–2156.3) | 1882.0 (1629.9–2124.3) | <0.001 | 41.0 (-144.6–207.9) |
| higher value | 1903.5 (1635.0–2208.0) | 1902.0 (1656.0–2207.0) | <0.001 | 51.5 (-131.5–225.8) |
| **Abdominal CT** | | | | |
| Subcutaneous fat area (cm$^2$) | 108.7 (69.7–162.7) | 101.0 (63.1–161.0) | <0.001 | -4.3 (-22.6–12.8) |
| Visceral fat area (cm$^2$) | 77.1 (41.9–120.4) | 74.7 (34.5–114.4) | 0.002 | -3.7 (-15.9–10.4) |

Δ, changes during 12 months; CTR, cardiothoracic ratio; HDL cholesterol, high-density lipoprotein cholesterol; LDL cholesterol, low-density lipoprotein cholesterol; CRP, C-reactive protein; hANP, human atrial natriuretic peptide; BNP, brain natriuretic peptide; LVEF, left ventricular ejection fraction; IVST, interventricular septum thickness; PWT, posterior left ventricular wall thickness; LVMI, left ventricular mass index; E/e', E over e-prime; Dec-T, deceleration time; ABI, ankle-brachial index; baPWV, brachial-ankle pulse wave velocity

significant difference in serum s(P)RR level was noted between patients in the death group and the survival group. Although the reason for this result is unclear, it may be possible that a 1-year observation period, which we set to estimate the association between serum s(P)RR level and changes in background factors without losing many patients due to death, was too short to investigate the relationship between serum s(P)RR level and prognosis. We are planning to assess the prognostic value of serum s(P)RR level after a longer follow-up period.

**Table 4. Comparison of changes (Δ) in each parameter between higher and lower groups of serum s(P)RR concentration in patients who survived.**

| | higher group | lower group | P |
|---|---|---|---|
| | [s(P)RR ≥29.8 ng/ml] | [s(P)RR <29.8 ng/ml] | |
| | (n = 117) | (n = 116) | |
| **Background factors** | | | |
| Body mass index (kg/m$^2$) | -0.1 (-0.6–0.4) | 0.0 (-0.6–0.5) | 0.656 |
| CTR (%) | 2.1 (0.1–4.3) | 1.3 (-0.8–3.4) | 0.098 |
| **Blood tests** | | | |
| Hemoglobin (g/dl) | -0.2 ± 1.5 | 0.3 ± 1.1 | <0.001 |
| HDL cholesterol (mg/dl) | -1.0 ± 9.0 | 1.0 (-3.0–5.0) | 0.124 |
| LDL cholesterol (mg/dl) | -2.0 (-15.0–11.5) | 1.3 ± 19.2 | 0.151 |
| Triglyceride (mg/dl) | -3.0 (-28.5–19.0) | 4.0 (-19.3–24.0) | 0.099 |
| Calcium (mg/dl) | 0.2 (-0.1–0.5) | 0.1 (-0.2–0.4) | 0.134 |
| Inorganic phosphorus (mg/dl) | -0.1 ± 1.7 | 0.2 ± 1.2 | 0.052 |
| Intact-Parathyroid hormone (pg/ml) | -1.0 (-46.0–52.0) | 2.0 (-41.8–50.0) | 0.957 |
| Creatinine (mg/dl) | 0.0 ± 1.5 | 0.6 (-0.2–1.3) | 0.003 |
| Uric acid (mg/dl) | -0.3 (-1.0–0.5) | 0.1 ± 0.9 | 0.016 |
| CRP (mg/dl) | 0.00 (-0.07–0.14) | 0.00 (-0.02–0.05) | 0.964 |
| Albumin (g/dl) | -0.1 (-0.2–0.1) | 0.0 (-0.1–0.1) | 0.093 |
| hANP (pg/ml) | 6.4 (-9.3–28.0) | -3.0 (-15.7–13.4) | 0.008 |
| BNP (pg/ml) | 20.0 (-34.9–113.6) | -4.7 (-73.3–50.8) | 0.035 |
| **Physical function tests** | | | |
| Echocardiography | | | |
| LVEF (%) | -1.9 (-7.2–3.6) | -1.2 (-5.5–3.7) | 0.509 |
| IVST (mm) | 0.8 ± 1.8 | 0.1 ± 1.4 | 0.005 |
| PWT (mm) | 0.5 ± 1.5 | 0.2 ± 1.3 | 0.130 |
| LVMI (g/m$^2$) | 7.1 (-18.5–37.7) | -1.1 (-20.8–36.8) | 0.567 |
| E/e' | 0.8 (-1.7–4.3) | 0.4 ± 4.7 | 0.295 |
| DecT | 8.8 (-37.9–37.9) | 4.4 ± 57.3 | 0.786 |
| ABI | | | |
| average value | 0.01 (-0.07–0.05) | -0.01 (-0.07–0.04) | 0.768 |
| lower value | -0.01 (-0.07–0.05) | 0.01 (-0.08–0.06) | 0.949 |
| baPWV (cm/s) | | | |
| average value | 50.5 (-118.4–236.0) | 36.0 (-144.9–195.2) | 0.997 |
| higher value | 45.5 (-159.8–243.8) | 52.0 (-124.5–203.0) | 0.740 |
| **Abdominal CT** | | | |
| Subcutaneous fat area (cm$^2$) | -2.4 (-28.8–14.7) | -6.5 (-21.8–9.6) | 0.538 |
| Visceral fat area (cm$^2$) | -6.1 ± 28.5 | -3.0 (-11.6–10.9) | 0.121 |

Δ, changes during 12 months; CTR, cardiothoracic ratio; HDL cholesterol, high-density lipoprotein cholesterol; LDL cholesterol, low-density lipoprotein cholesterol; CRP, C-reactive protein; hANP, human atrial natriuretic peptide; BNP, brain natriuretic peptide; LVEF, left ventricular ejection fraction; IVST, interventricular septum thickness; PWT, posterior left ventricular wall thickness; LVMI, left ventricular mass index; E/e', E over e-prime; Dec-T, deceleration time; ABI, ankle-brachial index; baPWV, brachial-ankle pulse wave velocity

In this study, the ratio of patients taking RAS-Is and CCBs was higher in the survival group than in the death group (Table 2). Because our sample size was relatively small and most patients were taking both RAS-Is and CCBs, statistical group comparisons between patients with and without antihypertensive drug therapy are less robust. Therefore, this issue should also be addressed by further investigations.

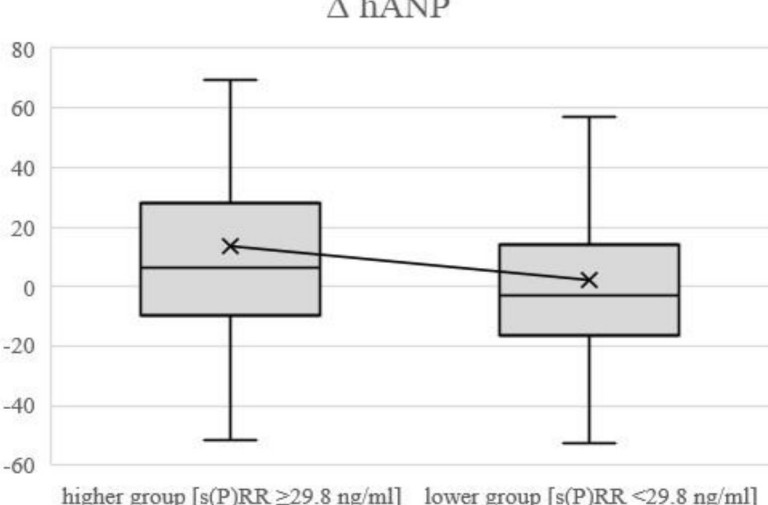

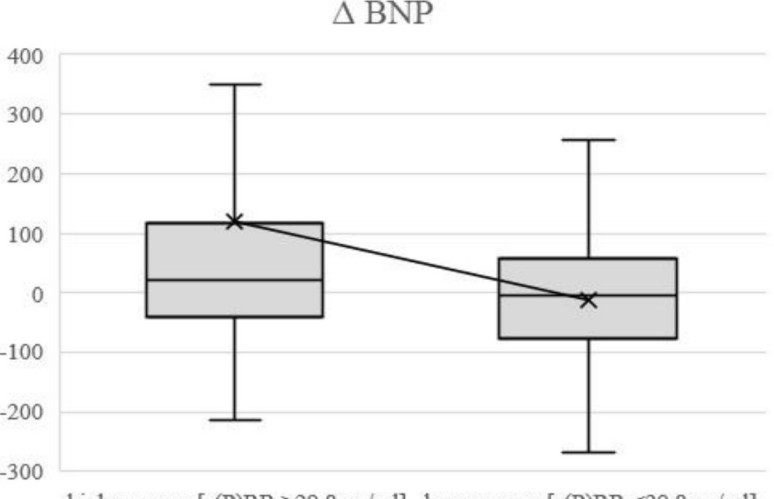

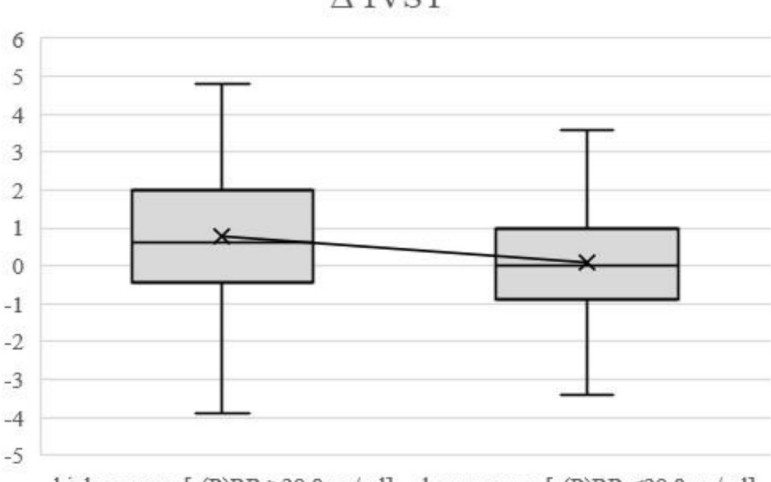

**Fig 1. Box plots comparing changes (Δ) in hANP, BNP, and IVST in the higher serum s(P)RR and lower serum s (P)RR groups.** Δ, changes during 12 months; hANP, human atrial natriuretic peptide; BNP, brain natriuretic peptide; IVST, intraventricular septum thickness; s(P)RR, soluble (pro)renin receptor.

**Table 5. Single correlation analyses with ΔBNP in patients who survived.**

| | ρ | P | | ρ | P |
|---|---|---|---|---|---|
| **Background factors** | | | BNP | -0.415 | <0.001 |
| Age | -0.010 | 0.906 | **Physical function tests** | | |
| Body mass index | -0.006 | 0.946 | Echocardiography | | |
| Duration of hemodialysis therapy | 0.045 | 0.585 | LVEF | 0.026 | 0.774 |
| CTR | -0.041 | 0.623 | IVST | 0.008 | 0.920 |
| **Blood tests** | | | PWT | -0.009 | 0.917 |
| s(P)RR | 0.132 | 0.109 | LVMI | -0.056 | 0.507 |
| Hemoglobin | 0.029 | 0.722 | E/e' | 0.022 | 0.796 |
| HDL cholesterol | -0.052 | 0.528 | Dec-T | -0.733 | 0.383 |
| LDL cholesterol | 0.199 | 0.015 | ABI | | |
| Triglyceride | 0.162 | 0.048 | average value | 0.009 | 0.919 |
| Calcium | -0.025 | 0.764 | lower value | 0.033 | 0.697 |
| Inorganic phosphorus | -0.068 | 0.408 | baPWV | | |
| Intact-Parathyroid hormone | 0.129 | 0.118 | average value | 0.104 | 0.258 |
| Creatinine | -0.119 | 0.150 | higher value | 0.119 | 0.194 |
| Uric acid | 0.003 | 0.971 | **Abdominal CT** | | |
| CRP | 0.028 | 0.737 | Subcutaneous fat area | 0.087 | 0.297 |
| Albumin | 0.014 | 0.866 | Visceral fat area | 0.060 | 0.471 |
| hANP | -0.304 | <0.001 | | | |

Δ, changes during 12 months; BNP, brain natriuretic peptide; CTR, cardiothoracic ratio; HDL cholesterol, high-density lipoprotein cholesterol; LDL cholesterol, low-density lipoprotein cholesterol; CRP, C-reactive protein; hANP, human atrial natriuretic peptide; LVEF, left ventricular ejection fraction; IVST, interventricular septum thickness; PWT, posterior left ventricular wall thickness; LVMI, left ventricular mass index; E/e', E over e-prime; Dec-T, deceleration time; ABI, ankle-brachial index; baPWV, brachial-ankle pulse wave velocity

## Comparison of changes (Δ) in each parameter between higher and lower groups of serum s(P)RR concentration

In our study, in patients who survived, ΔhANP and ΔBNP levels and ΔIVST were significantly higher in patients with serum s(P)RR ≥29.8 ng/ml than in those with s(P)RR <29.8 ng/ml (Table 4, Fig 1), suggesting that a status of higher serum s(P)RR is related to aggravation of cardiac hypertrophy and heart failure.

We have reported that the long-term administration of a (P)RR blocker attenuated the development of cardiac fibrosis and hypertrophy [28]. These findings raise the possibility that the activation of (P)RR in the heart may contribute to cardiac fibrosis and hypertrophy in HD patients, and further studies are needed to address this issue.

## Relationships between ΔBNP and serum s(P)RR levels

In our study, an association between high Δ BNP and the group with higher s(P)RR was observed even after correction for LDL-C, TG, hANP, and BNP, which were correlated with the ΔBNP level (Tables 5 and 6). These data indicate the possibility that an increased expression of (P)RR in the heart may cause worsening of heart failure independent of other Δ BNP-related factors at the start time of observation. They also suggest that serum s(P)RR concentration could be used as a biomarker for the progression of heart failure and thus may be useful in selecting patients who require more intensive care in terms of heart failure.

**Table 6. Multiple regression analyses with ΔBNP in patients who survived.**

| Variables | ΔBNP | |
|---|---|---|
| | β | p |
| s(P)RR | 0.159 | 0.042 |
| LDL cholesterol | - | - |
| Triglyceride | - | - |
| hANP | 0.163 | 0.057 |
| BNP | -0.611 | <0.001 |
| | $R^2 = 0.133$, P <0.001 | |
| | for entire model | |

Δ, changes during 12 months; BNP, brain natriuretic peptide; s(P)RR, soluble (pro)renin receptor; LDL cholesterol, low-density lipoprotein cholesterol; hANP, human atrial natriuretic peptide

## Limitations

We should acknowledge that there are some limitations to this study. Firstly, our sample size was relatively small. Secondly, the present data of HD patients may have been modulated by HD therapy, because s(P)RR is dialyzed to some extent[11]. Thirdly, the mechanisms by which serum s(P)RR level is associated with background factors remain unclear. Fourthly, the causal relationship between high serum s(P)RR level and high Δ BNP remains undetermined. Further studies are required to clarify the role of serum s(P)RR level in HD patients in more detail.

## Conclusions

High serum s(P)RR level in HD patients was associated with various background factors and several changes in cardiac function and atherogenic factors. Furthermore, high serum s(P)RR level was independently correlated with Δ BNP after correction for other Δ BNP-related factors. It may be possible that s(P)RR can be used as a marker of progression of heart failure in HD patients. Further studies are needed to determine the prognostic significance of serum s(P)RR concentration in HD patients and whether reducing serum s(P)RR level may improve the prognoses of these patients.

## Acknowledgments

We would like to thank Chikahito Suda, Noriko Morishima, and Chinami Muramatsu for their technical support for the s(P)RR assay. We also would like to thank Editage (www.editage.com) for English language editing.

## Author Contributions

**Conceptualization:** Yoshifumi Amari, Satoshi Morimoto.

**Data curation:** Yoshifumi Amari, Takeshi Iida, Hiroki Takimoto, Hidenobu Okuda, Takatomi Yurugi, Yasuo Oyama, Naoki Aoyama, Fumitaka Nakajima.

**Formal analysis:** Yoshifumi Amari.

**Investigation:** Yoshifumi Amari.

**Supervision:** Satoshi Morimoto, Atsuhiro Ichihara.

**Writing – original draft:** Yoshifumi Amari, Satoshi Morimoto.

**Writing – review & editing:** Yoshifumi Amari, Satoshi Morimoto.

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
