## [Decision Letter · Decision Letter 0]

7 Apr 2020

PONE-D-20-08545

Association between serum soluble (pro)renin receptor level and worsening of cardiac function in hemodialysis patients: A prospective observational study

PLOS ONE

Dear Dr. Morimoto,

Thank you for submitting your manuscript to PLOS ONE. After careful consideration, we feel that it has merit but does not fully meet PLOS ONE’s publication criteria as it currently stands. Therefore, we invite you to submit a revised version of the manuscript that addresses the points raised during the review process.

Two experts raised several concerns on your article.  Please revise to answer all the concerns and discuss the points.

We would appreciate receiving your revised manuscript by May 22 2020 11:59PM. To enhance the reproducibility of your results, we recommend that if applicable you deposit your laboratory protocols in protocols.io, where a protocol can be assigned its own identifier (DOI) such that it can be cited independently in the future. For instructions see: http://journals.plos.org/plosone/s/submission-guidelines#loc-laboratory-protocols

We look forward to receiving your revised manuscript.

Kind regards,

Tatsuo Shimosawa, M.D., Ph.D.

Academic Editor

PLOS ONE

Journal Requirements:

https://www.ncbi.nlm.nih.gov/pubmed/27367528

https://physoc.onlinelibrary.wiley.com/doi/10.14814/phy2.13587

https://journals.plos.org/plosone/article?id=10.1371/journal.pone.0195464

In your revision ensure you cite all your sources (including your own works), and quote or rephrase any duplicated text outside the methods section. Further consideration is dependent on these concerns being addressed.

Reviewers' comments:

Reviewer's Responses to Questions

**Comments to the Author**

1. Is the manuscript technically sound, and do the data support the conclusions?

Reviewer #1: Yes

Reviewer #2: Yes

2. Has the statistical analysis been performed appropriately and rigorously? 

Reviewer #1: Yes

Reviewer #2: Yes

3. Have the authors made all data underlying the findings in their manuscript fully available?

Reviewer #1: Yes

Reviewer #2: Yes

4. Is the manuscript presented in an intelligible fashion and written in standard English?

Reviewer #1: Yes

Reviewer #2: Yes

5. Review Comments to the Author

Reviewer #1: Morimoto et al. have investigated the relationship between sPRR and myocardial dysfunction in HD patients. The authors carefully analyzed the data and manuscript contains some novel information. However, there are several important concerns that should be addressed before resubmitting revised manuscript. The specific points are below.

Critiques:

1. This reviewer strongly recommends to reconstruct the Introduction. First of all, any general knowledge about PRR and HD/CVD should be removed. Then, the authors should clearly mention what is known and what is not known about sPRR, especially for CKD/CVD. As you know, there are many reports that sPRR is a good biomarker for CKD/CVD, except references #11 and 30. Finally, rational of this study and specific hypothesis should be mentioned.

2. Similarly, discussion should focus on the obtained data. Any introduction and general knowledge about PRR and HD/CVD should be removed.

3. The authors should discuss why basal sPRR levels are different among the reports.

4. Is there any difference between diabetic and nondiabetic patients?

Reviewer #2: The authors investigated the role of s(P)RR on cardiac function in hemodialysis patients. The study is interesting, and the manuscript is easy to read. There are several comments which could improve the conclusions of this manuscript.

First, the mechanistic explanation of s(P)RR in hemodialysis is missing. Where does the blood s(P)RR is secreted in hemodialysis patients? Second, the background of the medication should be provided and discuss whether these have influenced to the results. Third, the association plot between delta BNP and s(P)RR should be figured. Finally, did hemodialysis parameters such as dry weight, Qb, Qd and types of dialyzer influenced the results. Please provide and discuss.

6. PLOS authors have the option to publish the peer review history of their article (what does this mean?). If published, this will include your full peer review and any attached files.

Reviewer #1: No

Reviewer #2: No

---

## [Author Response · Author response to Decision Letter 0]

23 Apr 2020

Comments to Reviewer #1:

Thank you very much for your excellent comments.

1)Your comment: 

This reviewer strongly recommends to reconstruct the Introduction. First of all, any general knowledge about PRR and HD/CVD should be removed. Then, the authors should clearly mention what is known and what is not known about sPRR, especially for CKD/CVD. As you know, there are many reports that sPRR is a good biomarker for CKD/CVD, except references #11 and 30. Finally, rational of this study and specific hypothesis should be mentioned.

>>> Our comment: 

Following your suggestion, we revised the Introduction by removing general knowledge about (P)RR and HD/CVD, by describing what is known and unknow about s(P)RR in HD patients, and by showing our hypothesis.

2) Your comment:

Similarly, discussion should focus on the obtained data. Any introduction and general knowledge about PRR and HD/CVD should be removed.

>>> Our comment: 

Following your suggestion, we removed parts describing general knowledge about PRR and HD/CVD which reads: “Systemic inflammation is common in HD patients, and CRP is released from hepatocytes in response to inflammation. Elevated serum CRP is a strong predictor of mortality, and higher CRP levels are associated with worse outcomes in HD patients. Protein energy wasting is a state of decreased body stores of protein and energy fuel and is associated with diminished functional capacity, impaired quality of life, and increased morbidity and mortality in HD patients. It has also been reported that malnutrition is the most important factor associated with increased mortality in HD patients. Although it remains unclear whether malnutrition directly causes mortality or is a result of diseases that cause mortality” (L.4 to L.12, P.24 in the Revised Manuscript with Track Changes), “A previous report demonstrated that gene expressions of (P)RR, renin, and angiotensinogen were elevated in the heart and kidneys of rats with chronic heart failure. It has been reported that patients with heart failure had significantly higher plasma s(P)RR levels than control subjects. Local (P)RR gene delivery into the heart showed deleterious effects on cardiac function via activation of angiotensin II-independent extracellular matrix remodeling. Cardiac-specific overexpression of (P)RR was shown to cause both electronic remodeling and structural remodeling in the atria by activation of the angiotensin II-independent pathway and eventually led to atrial fibrillation. High salt intake enhanced the cardiac expressions of (P)RR, leading to the acceleration of cardiac interstitial fibrosis, perivascular fibrosis, and cardiomyocyte hypertrophy via angiotensin II-dependent and -independent pathways at an early stage of hypertension” (L.14, P.26 to L.8, P.27 in the Revised Manuscript with Track Changes), and “There are several reports demonstrating that BNP level is particularly useful for diagnosing heart failure, evaluating heart failure severity, and predicting cardiovascular events and risk of death in HD patients” (L.15 to 17, P. 27 in the Revised Manuscript with Track Changes).

3) Your comment:

The authors should discuss why basal sPRR levels are different among the reports.

>>> Our comment: 

Thank you very much for your valuable comment. Following your suggestion, we discussed why basal s(P)RR levels are different among the reports as follows: “It has been shown that basal blood s(P)RR levels differ among reports with a tendency of blood s(P)RR levels to be high in older subjects [4, 20], patients with heart failure [7], renal dysfunction [21], or Graves’ disease [22], and pregnant women [23, 24]. The reason of the alterations of these values may be due to the differences of the disease and conditions of the subjects [24]. Our study subjects undergoing maintenance HD also showed high serum s(P)RR levels, which were significantly higher when compared with subjects with normal renal function [8]. Although the reason for this remains unclear, there are several possible explanations. First, elevated serum s(P)RR levels may reflect an increase in expression of (P)RR in the kidneys. Second, increased expression of (P)RR in other organs such as the heart and brain, where the expression levels of (P)RR are extremely high [1], may be the cause of elevated serum s(P)RR level. Third, elevated serum s(P)RR levels may be simply due to a decrease or loss of excretion from the kidney” (L. 4 to 16, P.22 in the Revised Manuscript with Track Changes).

4) Your comment:

Is there any difference between diabetic and nondiabetic patients?

>>> Our comment: 

Thank you very much for excellent comment. We added data about diabetes in Tables 1 and 2. We also revised the Results as follows: “Primary disease (diabetic or not) was not significantly different between the 2 groups” (L.8 to 9, P.11 in the Revised Manuscript with Track Changes) and “No significant differences in primary disease (DM/no DM) was noted between in the survival and death groups” (L.1 to 2, P.14 in the Revised Manuscript with Track Changes).

 

Comments to Reviewer #2:

Thank you very much for your excellent comments.

1)Your comment: 

The mechanistic explanation of s(P)RR in hemodialysis is missing. Where does the blood s(P)RR is secreted in hemodialysis patients? 

>>> Our comment: 

Thank you very much for your valuable comment. We discussed why serum s(P)RR levels are increased in HD patients as follows: “Our study subjects undergoing maintenance HD also showed high serum s(P)RR levels, which were significantly higher when compared with subjects with normal renal function [8]. Although the reason for this remains unclear, there are several possible explanations. First, elevated serum s(P)RR levels may reflect an increase in expression of (P)RR in the kidneys. Second, increased expression of (P)RR in other organs such as the heart and brain, where the expression levels of (P)RR are extremely high [1], may be the cause of elevated serum s(P)RR level. Third, elevated serum s(P)RR levels may be simply due to a decrease or loss of excretion from the kidney” (L.8 to 16, P.22 in the Revised Manuscript with Track Changes).

2)Your comment:

The background of the medication should be provided and discuss whether these have influenced to the results. 

>>> Our comment: 

Thank you for your excellent comment. Following your suggestion, we added data of selected medication in Tables 1, 2, and 5. We found that the selected medication influenced the data of serum s(P)RR and prognosis but not ΔBNP, the most important part of this study. We revised the Materials and Methods as follows: “At the start of this study, we collected information on the study population, including age, sex, body mass index (BMI), primary disease (diabetic or not), duration of HD, smoking status, and selected medication” (L.9 to 11, P.6 in the Revised Manuscript with Track Changes). And we corrected Results as follows: “With respect to antihypertensive treatment, the ratio of patients taking RAS-inhibitors (RAS-Is), such as angiotensin-converting enzyme inhibitors, angiotensin receptor blockers (ARBs), and renin inhibitors, was significantly higher in the lower serum s(P)RR group (83.7 %) than in the higher serum s(P)RR group (69.0 %)” (L.4 to 8, P.11 in the Revised Manuscript with Track Changes), and “With respect to antihypertensive treatment, the ratio of taking RAS-Is and calcium channel blockers (CCBs) were significantly higher in the survival group (79.4 % and 77.3 %, respectively) than the death group (45.8 % and 41.7 %, respectively)” (L.15 to 18, P.13 in the Revised Manuscript with Track Changes), and “The ratio of patients taking RAS-Is, β-blockers, and CCBs did not show significant relationship with ΔBNP” (L.2 to 5, P.14 in the Revised Manuscript with Track Changes). In addition, we revised Discussion: “In our study, the patients in the lower serum s(P)RR group took RAS-Is more frequently than those in the higher serum s(P)RR group (Table 1). It has been reported that a significantly lower level of serum s(P)RR was observed in CKD patients treated with ARBs [21], consistent with our study” (L.16, P. 23 to L. 1, P. 24 in the Revised Manuscript with Track Changes), and “In this study, the ratio of patients taking RAS-Is and CCBs was higher in the survival group than in the death group (Table 2). Because our sample size was relatively small and most patients were taking both RAS-Is and CCBs, statistical group comparisons between patients with and without antihypertensive drug therapy are less robust. Therefore, this issue should also be addressed by further investigations” (L.4 to 8, P.26 in the Revised Manuscript with Track Changes).

3)Your comment:

The association plot between delta BNP and s(P)RR should be figured. 

>>> Our comment: 

Following your suggestion, we made an association plot between serum s(P)RR level and ΔBNP (Supplemental Figure 1). In line with the result showing no significant relationship between serum s(P)RR level and ΔBNP as a whole (Table 5), the figure was not good enough to show a strong relationship between serum s(P)RR level and ΔBNP. However, when the subjects were divided into the higher and lower s(P)RR groups, ΔBNP was significantly higher in the higher s(P)RR group than in the lower s(P)RR group as already shown in Table 4).” We added this explanation in Results as follows instead of showing the Supplemental Figure 1: “Serum s(P)RR level was not associated with ΔBNP level (Table 5). On the other hand, when the subjects were divided into the higher and lower s(P)RR groups, ΔBNP was significantly higher in the higher s(P)RR group than in the lower s(P)RR group (Figure 1)” (L.11 to 14, P.19 in the Revised Manuscript with Track Changes).

3)Your comment:

Did hemodialysis parameters such as dry weight, Qb, Qd and types of dialyzer influenced the results. Please provide and discuss. 

>>> Our comment: 

Thank you very much for your valuable comment. As an index of body weight, we assessed BMI instead of dry weight which may not reflect the actual body weight and we found that BMI was not associated with serum s(P)RR level at baseline (Table 1). Qb, Qd, and types of dialyzer may affect the hemodialysis efficiency. We assessed Kt/V instead of Qb, Qd, and types of dialyzer, because Kt/V is considered to be a better index of hemodialysis efficiency. And as a result, we found that Kt/V was not significantly different between higher and lower groups of serum s(P)RR concentration (revised Table 1) and between survival and death groups (revised Table 2) and that there was no relationship between Kt/V and ΔBNP level (revised Table 5).

We added data of Kt/V in Tables 1, 2, and 5. In addition, we revised the Materials and Methods as follows: “The Kt/V was calculated on the 1st dialysis day of the week using the following equation, the formula of Daugirdas [9]: Kt/V = - Ln [post-dialysis value of BUN / pre-dialysis value of BUN - 0.008 x dialysis time] + (4 - 3.5 x post-dialysis value of BUN / pre-dialysis value of BUN) x amount of drainage / post-dialysis body weight}” (L.14 to 17, P. 6 in the Revised Manuscript with Track Changes). In addition, we revised the Results as follows: “In addition, neither BMI nor Kt/V was significantly different between the 2 groups, suggesting that hemodialysis parameters may not have strong influences on the serum s(P)RR levels” (L.9 to 11, P.11 in the Revised Manuscript with Track Changes).

---

## [Decision Letter · Decision Letter 1]

4 May 2020

Association between serum soluble (pro)renin receptor level and worsening of cardiac function in hemodialysis patients: A prospective observational study

PONE-D-20-08545R1

Dear Dr. Morimoto,

We are pleased to inform you that your manuscript has been judged scientifically suitable for publication and will be formally accepted for publication once it complies with all outstanding technical requirements.

With kind regards,

Tatsuo Shimosawa, M.D., Ph.D.

Academic Editor

PLOS ONE

Additional Editor Comments (optional):

Reviewers' comments:

Reviewer's Responses to Questions

**Comments to the Author**

1. If the authors have adequately addressed your comments raised in a previous round of review and you feel that this manuscript is now acceptable for publication, you may indicate that here to bypass the “Comments to the Author” section, enter your conflict of interest statement in the “Confidential to Editor” section, and submit your "Accept" recommendation.

Reviewer #1: All comments have been addressed

2. Is the manuscript technically sound, and do the data support the conclusions?

Reviewer #1: Yes

3. Has the statistical analysis been performed appropriately and rigorously? 

Reviewer #1: Yes

4. Have the authors made all data underlying the findings in their manuscript fully available?

Reviewer #1: Yes

5. Is the manuscript presented in an intelligible fashion and written in standard English?

Reviewer #1: Yes

6. Review Comments to the Author

Reviewer #1: The authors responded to my all comments and manuscript is now much improved. I have no further comment.

7. PLOS authors have the option to publish the peer review history of their article (what does this mean?). If published, this will include your full peer review and any attached files.

Reviewer #1: No

---

## [Editor Report · Acceptance letter]

21 May 2020

PONE-D-20-08545R1 

Association between serum soluble (pro)renin receptor level and worsening of cardiac function in hemodialysis patients: A prospective observational study 

Dear Dr. Morimoto:

I am pleased to inform you that your manuscript has been deemed suitable for publication in PLOS ONE. Congratulations! Your manuscript is now with our production department. 

With kind regards,

on behalf of

Prof. Tatsuo Shimosawa 

Academic Editor

PLOS ONE